# The influence of 4-week eccentric Nordic hamstring exercise training on postural balance and muscle strength: A randomized controlled trial

Magdalena Podczarska-Głowacka[1]*, Ewelina Perzanowska[1], Katarzyna Krasowska[1], Zuzanna Trapik[1], Agata Kalkowska[1], Sebastian Klich[2]

1 Department of Basic Physiotherapy, Gdansk University of Physical Education and Sport, Gdansk, Poland, 2 Department of Sport Didactics, Wrocław University of Health and Sport Sciences, Wrocław, Poland

* magdalena.podczarska-glowacka@awf.gda.pl

## Abstract

This study aimed to determine the impact of a 4-week Nordic Hamstrings Exercise (NHE) eccentric training program on postural balance in static and dynamic conditions, focusing on the dominant leg. This study involved thirty-two recreationally active individuals, randomly divided into two groups: Nordic Exercise training (NHE Group; n = 16) and the matched-control group (CON Group; n = 16). The NHE Group performed Nordic hamstring exercise training for 4 weeks, during which the volunteer exercised 3 times a week for 3 sets, each set consisting of 3 repetitions of NHE. Postural balance was assessed using three tests: static balance on a stable platform with eyes closed, dynamic balance on an unstable platform for both legs and dynamic balance on an unstable platform for the dominant leg. Three indices were measured using the Biodex Stability System: Anterior-Posterior Stability Index (APSI), Medio-Lateral Stability Index (MLSI), and Overall Stability Index 1 (OSI). The NHE group significantly improved balance on a stable platform with eyes closed compared to baseline ($p \leq 0.001$) and on unstable platforms compared to both baseline and the control group ($p \leq 0.001$). Both groups improved knee strength, but the NHE group showed significantly greater improvements ($p \leq 0.001$ for NHE, $p \leq 0.04$ for control). Strong positive correlations ($r = 0.5$ to $0.7$) were found between stability measures, suggesting shared underlying mechanisms influencing balance ($p < 0.01$). A 4-week NHE training protocol effectively improves postural balance and performance in athletes. Measuring postural control is valuable for assessing neuromuscular function and injury prevention in athletic populations. This study suggests that eccentric NHE exercises may improve athlete balance, reduce hamstring injury risk, and decrease the need for rehabilitation by activating posterior thigh muscles.

**Data availability statement:** All files are available from the database Podczarska-Głowacka, Magdalena, 2024, "Nordic hamstring exercise, muscle strength and postural balance", https://doi.org/10.18150/I68R9W, RepOD, V1.

**Funding:** The author(s) received no specific funding for this work.

**Competing interests:** The authors have declared that no competing interests exist.

## Introduction

Hamstring injuries are common among professional and recreational athletes [1]. Specifically, these injuries tend to recur, creating a complex issue for coaches and the medical team [2]. In many cases, a hamstring injury results in significant time loss in training [3] and reduced recreational physical activity [4]. Fasuyi et al. reported that hamstring muscle imbalance causes poor posture during static or dynamic movements [5]. Nordic hamstring exercises (NHE) are eccentric-type exercises that improve muscle strength and activation muscle. Eccentric strengthening of the knee flexors and efficient dynamic balance can improve motor performance and reduce the risk of injury in athletes [6,7].

The NHE has been utilized in both recreational and elite sports training (e.g., football, handball, rugby, bodybuilding) as a method to prevent injuries to the knee joint and hamstring muscles [8–12]. Saleh et al. found that NHE can lead to significant improvements in dynamic balance among athletes. Greater balance ability may reduce the risk of injury, thereby lowering the costs associated with treatment and rehabilitation. Furthermore, Saleh et al. [7] demonstrated that six weeks of NHE training, combined with the Copenhagen addition exercise, improved balance on an unstable platform (levels 1–12). Alt et al. conducted a training program based on four weeks of NHE focused on thigh muscle strength, balance, and kinematic analysis [13], while Vianna et al. implemented eight weeks of NHE training without assessing balance [14]. The connection between NHE and postural balance is further supported by research indicating that NHE can significantly enhance dynamic balance performance among athletes. A study found that participants who performed NHE demonstrated improved balance capabilities, which are essential for reducing injury risks and enhancing overall motor performance in sports settings [15].

The NHE training procedure with assessment of postural balance for static and dynamic platforms has not been sufficiently researched, despite the use of eccentric hamstring exercises. Various measurements were used to determine dynamic balance. Khan et al. assessed dynamic balance by the star excursion balance test (SEBT) [16]. However, Encarnación-Martínez et al. used the modified mSEBT and Dynamic Postural Stability Index (DPSI) [17]. Compared to other methods, the Biodex Balance System appears to be a more advantageous option for simultaneous assessment of postural balance in real-time with high validity and reliability for both static and dynamic platforms. Therefore, the study used the Biodex Balance System to assess balance in three types of Postural Stability: standing on both legs on a stable platform with closed eyes; standing on both legs on an unstable platform using stiffness level 5; and for the dominant leg on an unstable platform using stiffness level 4.

A decreased level of muscular strength and postural balance has been related to a higher risk of hamstring muscle injuries [1,2]. From a biomechanical perspective, assessing the interaction between muscular strength, balance, and injury risk is complex. These factors are interconnected and influenced by various biomechanical and neuromuscular properties of the musculoskeletal system [3,4]. Previous studies investigated limit of stability after 6 weeks of NHE training using the Biodex Stability System [3], while other studies assessed dynamic balance after both single [5]

and 6 weeks of NHE training using the Y-Balance Test. To our knowledge, the effectiveness of NHE training on postural balance remains unclear. Therefore, this study aimed to address the limitations of previous studies [3–5] by comprehensively assessing postural balance across three planes of motion (anterior-posterior, medial-lateral, and overall stability index ((APSI, MLSI, and OSI)), providing a more holistic understanding of the impact of NHE on balance. This innovative approach, by examining balance across multiple planes, may lead to a more nuanced understanding of the specific mechanisms by which NHE training improves balance and its implications for injury prevention. This randomized controlled trial (RCT) study aimed to investigate the primary effect of a 4-week NHE eccentric training program on improving dynamic balance compared to a control group among recreationally active individuals. Furthermore, secondary outcomes included changes in muscle strength. We hypothesized that a 4-week NHE eccentric training program will significantly improve dynamic balance in recreationally active individuals compared to a control group. This improvement will be observed across all three planes of postural stability (APSI, MLSI, and OSI) and in hamstring muscle strength.

## Materials and methods

### Study design

This study employed a randomized, controlled, single-blind design with a repeated-measures approach and was conducted between July 2024 and August 2024 in the Physical Exercise Laboratory at AWFiS in Gdańsk. To ensure rigorous reporting, we adhered to the Consolidated Standards of Reporting Trials (CONSORT) guidelines for pragmatic trials. Group allocation was randomized, resulting in two equal groups of 16 participants, including Nordic Exercise training (NHE) and a control (CON) group. This study investigated the effect of 4-weeks of NHE training on the ability to maintain balance. The Postural Stability Test was evaluated with a Biodex Balance System SD 115VAC (BBS). The measurement was performed at three time points: (1) at baseline, (2) after 4 weeks of NHE training, and (3) after a follow-up at 8 weeks (Figs 1, 2). Postural stability control was performed in three tests: (1) on a stable platform for both legs with closed eyes, (2) on an unstable platform for both legs using stiffness 5 levels (12 as the most stable platform, 1 as the smallest), and (3) for the dominant leg on an unstable platform using stiffness level 4. The concentric peak torque of the knee flexors and extensors was measured by a Biodex dynamometer (Biodex System 4 Pro, Biodex Medical Systems, Inc.). The study was approved by the Bioethics Committee at the Gdańsk University of Physical Education and Sport (resolution No. 1, approval date: February 1, 2024). The authors confirm that all ongoing and related trials for this intervention are registered. The trial was retrospectively registered on ClinicalTrials.gov (NCT06382597), with record verification approved in April 2024. Participant recruitment began before registration due to an oversight during the initial planning phase, where the importance of prospective registration was not fully appreciated. Upon recognizing this issue, the trial was promptly registered to ensure compliance with ethical standards and transparency. The authors confirm that all ongoing and related trials for this intervention are now registered prospectively. All participants read the form and provided written informed consent to participate in this study. Moreover, this study was conducted following the Declaration of Helsinki.

### Participants

The study involved thirty-two (n = 32) recreationally active individuals screened for eligibility from May 2024 to June 2024. The subjects were divided into two groups: the intervention group performing Nordic Exercise training (NHE Group; n = 16); 22.2 ± 1.4 yrs, 172.9 ± 8.8 cm, 72.9 ± 17.7 kg and the matched-control group (CON Group; n = 16); 21.7 ± 2.2 yrs, 171.9 ± 7.9 cm, 69.8 ± 13.5 kg (Table 1). All participants were students of the Gdansk University of Physical Education and Sport. Inclusion criteria included training experience >5 years in soccer and training duration >8 h per week for the past 6 months. The exclusion criteria included knee, hip, or lower back pain, injury, or surgery within 6 months before the intervention.

The recruitment process consisted of an interview during which questions were asked about physical activity, training experience, and weekly duration of activity. Participants were randomly divided into two groups, i.e.,: NHE and CON.

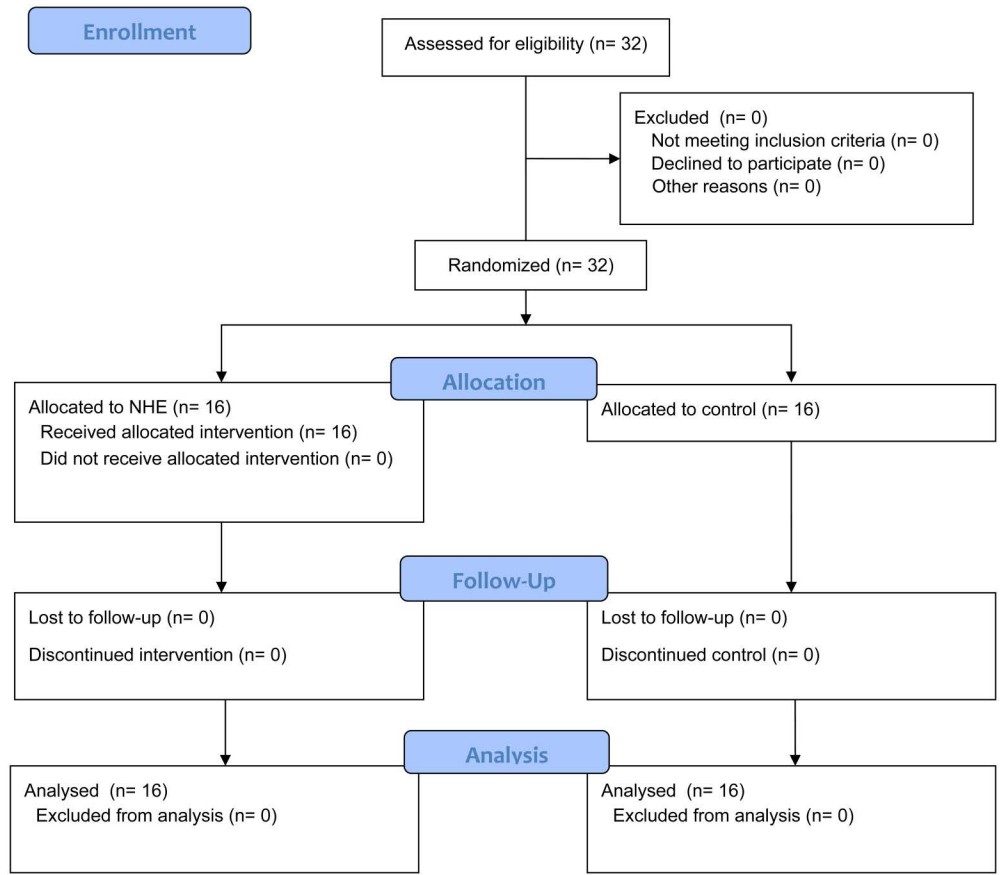

**Fig 1. Enrolment, randomization, and drop out of participants allocated to the intervention group (NHE) and matched-control group (CON).**

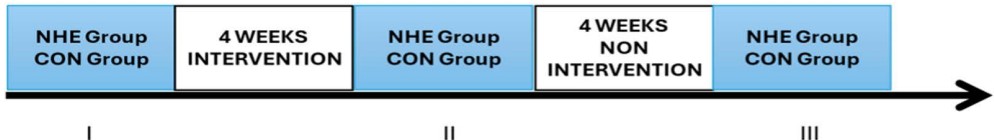

**Fig 2. The experimental procedure includes measurements taken at baseline (I), after 4 weeks (II), and a follow up at 8 week (III) from the start of the procedure.**

**Table 1. Subject's characteristics (mean ± SD).**

|  | NHE Group (n = 16) | CON Group (n = 16) | Total (n = 32) |
|---|---|---|---|
| **Age** | 22.2 ± 1.4 | 21.7 ± 2.2 | 21.9 ± 1.5 |
| **Height (cm)** | 172.9 ± 8.8 | 171.9 ± 7.9 | 172.5 ± 3.4 |
| **Weight (kg)** | 72.9 ± 17.7 | 69.8 ± 13.5 | 71.4 ± 15.6 |

Allocation was determined using a random number generator (random.org). Each participant was assigned a unique number, corresponding to a sealed envelope containing their group assignment. An independent researcher, blinded to the group assignments, selected these envelopes. Participants learned their group assignment only after completing baseline data collection. This information remained confidential from the researcher conducting the study.

The sample size effect was calculated by using G*Power software (version 3.1.9.2; Kiel University, Kiel, Germany). We calculated the power (1-β) for ANOVA within and between factors, including number of groups (2) and number of measurements (3). A sample size was estimated for a total of 32 (for all balance and strength tests), set a minimum expected effect size (Cohen's f) of 0.3, an α level of 0.05, and a power of 0.95 and correlation of 0.5.

### Experimental procedures

A week before the start of the experimental procedure, participants were invited to the Physical Exercise Laboratory of AWFiS in Gdańsk to familiarize themselves with the Nordic Hamstring exercise procedure and all outcome devices. All participants were familiarized with the research procedure and all test procedures during the measurement conducted 7 days before the main test. During the experiment, all participants completed a standardized warm-up routine. This included a warm-up on a cycle ergometer for 15 minutes, at 40 W and a speed of 20 km/h [11]. After completing baseline measurements, the NHE group received the intervention, while the CON group received no intervention. The NHE Group performed Nordic hamstring exercise training for 4 weeks, during which the volunteer exercised 3 times a week for 3 sets, each set consisting of 3 repetitions of NHE. After each series, there was a 2-minute passive break, during which the subject rested. In the first week, the subject exercised under the supervision of the trainer, in the next three weeks the subject exercised in training/home conditions after prior instruction. Nordic hamstring exercises consisted of a controlled fall of the straightened torso from a vertical position in a kneeling position forward towards the ground (parallel to the ground) and return to the starting position. The ankle joints were stabilized by another person's grip during a series of exercises. If the tested person had a problem with maintaining the trunk in conditions of controlled eccentric work, then the upper limbs protected and cushioned the contact with the floor, followed by a rebound from the hand to the starting position. The CON Group performed training activities as before, including traditional exercises to strengthen the hamstring muscles.

### Measurements outcomes

**Postural stability test.** In this study, balance ability was investigated using the Biodex balance system (BBS). The postural stability was assessed by performing a test on each participant using a Biodex Balance System (SD 115VAC, USA). The following measure parameters were used, i.e.,: Anterior-Posterior Stability Index (APSI), Medio-Lateral Stability Index (MLSI), and Overall Stability Index (OSI). Next, all parameters were calculated by the platform's software (Balance System SD Software Update v4.0.18, USA) based on the degree of the platform's oscillation; lower values reflected better postural control in the subjects. The subject's index was defined as an average of the three trials. PST reported the ability to maintain postural stability. PST assessed the ability to maintain postural stability. The subject's score was inversely proportional to the number of deviations from the center; thus, lower PST scores indicated better performance. The subject's task was to maintain the marker as close to the center of the on-screen disk as possible. Each static platform measurement consisted of three attempts, separated by 20-second intervals and 10-second breaks. The trial was invalid if the participant removed the foot from the platform, lost balance, or leaned on the railing. Before the test, the participant stood on a stationary platform without shoes and with his feet freely apart. Demographic data were collected, including age, height, and foot morphology (metatarsal and calcaneal position). Each participant completed three Postural Stability Tests:

1. Eyes-closed, bipedal stance on a stable surface.

2. Bipedal stance on an unstable surface with varying levels of stiffness (12−1, with 12 being the most stable),

3. For the dominant leg on an unstable platform use stiffness level 4.

**Isokinetic knee extension/flexion.**  All tests were performed using a calibrated Biodex System 4 Pro dynamometer with a standard knee unit attachment. To warm up and familiarize themselves with the lever arm movement, subjects performed 3 maximal concentric contractions of knee extension and flexion at an angular velocity of 240°/s, with 30-second rest intervals between each contraction. The highest peak torque of the 3 maximal contractions at each angular velocity and for each muscle group was obtained from the comprehensive data sheet and used as the criterion score. Participants were provided with real-time visual feedback of torque output and standardized verbal reinforcement during maximal voluntary contractions.

## Statistical analysis

The data and statistical analysis were performed using SPSS statistical software (version 24, SPSS Inc., Chicago, Illinois, USA). Mean values ± standard deviation (SD) or confidence interval (CI 95%) are reported. Normality of data distribution was assessed using the Shapiro-Wilk test, while homogeneity of variance was evaluated with Levene's test.

A two-way 3 x 2 mixed-model analysis of variance (ANOVA) with *Time* (baseline, after 4 weeks, and after 8 weeks) and *Intervention* (NHE group and CON group) was analyzed for the Postural Stability Test. If a significant interaction between factors was found, the Bonferroni adjustment for multiple comparisons was used for post-hoc tests ($p = 0.005$). Correlation coefficients were classified as trivial (0.0), small (0.1), moderate (0.3), strong (0.5), very strong (0.7), nearly perfect (0.9), and perfect (1.0) [18]. For all statistical tests, a p-value ≤ 0.05 was considered significant.

## Results

### Anthropometric data

The study involved thirty-two (n = 32) participants aged 20–26 (men (n = 13) and women (n = 19)); 21.9 ± 1.5 yrs, 172.5 ± 3.4 cm, 71.4 ± 15.6 kg) (Table 1).

### Stable platform with closed eyes

Two-way mixed-model ANOVAs revealed statistically significant main effects of *Time* ($F_{(2,60)} = 8.8$, $p = 0.001$, $\eta^2 = 0.22$) for Overall Stability Index (OSI), Anterior-Posterior Stability Index (APSI) ($F_{(2,60)} = 5.8$, $p = 0.001$, $\eta^2 = 0.16$), and Medio-Lateral Stability Index (MLSI) ($F_{(2,60)} = 6.2$, $p = 0.001$, $\eta^2 = 0.17$). In the NHE Group, all variables (OSI, APSI, and MLSI) on the stability platform with eyes closed decreased significantly 4 weeks ($p \leq 0.001$) and 8 weeks ($p \leq 0.001$) after the study for OSI and APSI compared to baseline. No significant differences were observed between the 4-week and 8-week assessments. In contrast, the CON Group showed no statistically significant differences in any index compared to baseline measurements (Table 2).

Table 2.  Stable platform with closed eyes.

| Variables | Nordic Exercise training group | | | | | | | | Control group | | | | | | | |
|---|---|---|---|---|---|---|---|---|---|---|---|---|---|---|---|---|
| | Baseline | After 4-week | After 8-week | Δ [4-week −baseline] | p*-value | Δ [8-week − 4-week] | p**-value | p***-value | Baseline | After 4-week | After 8-week | Δ [4-week − base-line] | p*-value | Δ [8-week − 4-week] | p**-value | p***-value |
| OSI | 0.48±0.24 | 0.21±0.08 | 0.26±0.11 | 0.27 | 0.001 | 0.05 | 0.001 | 1.0 | 0.32±0.13 | 0.34±0.12 | 0.35±0.23 | 0.02 | 1.0 | 0.01 | 1.0 | 1.0 |
| APSI | 0.34±0.14 | 0.19±0.05 | 0.16±0.06 | 0.15 | 0.001 | −0.03 | 0.001 | 1.0 | 0.24±0.07 | 0.21±0.07 | 0.26±0.24 | −0.03 | 1.0 | 0.05 | 1.0 | 1.0 |
| MLSI | 0.28±0.29 | 0.12±0.07 | 0.16±0.11 | 0.16 | 0.001 | 0.04 | 0.01 | 1.0 | 0.17±0.09 | 0.17±0.09 | 0.14±0.04 | 0.0 | 1.0 | −0.03 | 1.0 | 1.0 |

Significant differences *- within-group differences between 4 weeks after exercise and baseline, **- within-group differences between 8 weeks after (4 weeks after intervention) and baseline; ***- between-group differences between 8 weeks and 4 weeks; (p ≤ 0.05).

## Instability platform in the range of 5 mobility levels

Table 3 shows the mean ± SD of three stability indicators (OSI, APSI, and MLSI) at baseline, 4 weeks, and 8 weeks after NHE training. Two-way mixed-model ANOVAs revealed significant main effects of *Time* for OSI ($F_{(2,60)} = 28$, $p < 0.001$, $\eta^2 = 0.48$), APSI ($F_{(2,60)} = 23.7$, $p < 0.001$, $\eta^2 = 0.44$), and MLSI ($F_{(2,60)} = 7.9$, $p < 0.001$, $\eta^2 = 0.2$). In the NHE group, OSI and APSI significantly decreased after 4 weeks ($p \leq 0.001$) and 8 weeks (both $p \leq 0.001$) compared to baseline. The CON group showed a non-significant trend towards improvement in most indicators. To compare groups, stability indices within a specific range (e.g., 5) were analyzed. At 4 and 8 weeks, the NHE group exhibited non-significantly better OSI ($p = 0.03$) and APSI ($p = 0.04$) compared to the CON group (Table 5).

## Dominant leg on an unstable platform in the range 4 platform mobility level

Two-way mixed-model ANOVAs revealed statistically significant main effects of *Time* for OSI ($F_{(2,60)} = 35.8$, $p < 0.001$, $\eta^2 = 0.5$), APSI ($F_{(2,60)} = 26.1$, $p < 0.001$, $\eta^2 = 0.46$), and MLSI ($F_{(2,60)} = 29.7$, $p < 0.001$, $\eta^2 = 0.5$). Post hoc analysis revealed that in the NHE group, OSI, APSI, and MLSI decreased significantly at 4 weeks and 8 weeks compared to baseline ($p \leq 0.001$). In contrast, the CON group showed a non-significant changes in MLSI at 8 weeks compared to baseline ($p = 0.04$) (Table 4). At 4 weeks, the NHE group showed non-significantly better OSI ($p = 0.02$), APSI ($p = 0.04$), and MLSI ($p = 0.04$) compared to the CON group (Table 5).

## Concentric knee extension and flexion for right leg and left leg

Two-way mixed-model ANOVAs revealed statistically significant main effects of *Time* for right leg knee flexors ($F_{(2,60)} = 32.9$, $p < 0.001$, $\eta^2 = 0.52$) and extensors ($F_{(2,60)} = 28.3$, $p < 0.001$, $\eta^2 = 0.48$). Post hoc analyses showed that concentric knee flexor and extensor forces significantly increased at 4 and 8 weeks compared to baseline in the NHE group ($p \leq 0.001$). In the CON group, significant improvements were observed at 4 and 8 weeks for knee flexors ($p \leq 0.001$).

**Table 3. Unstable platform in the range of 5 mobility levels.**

| Vari-ables | Nordic Exercise training group | | | | | | | | Control group | | | | | | | |
|---|---|---|---|---|---|---|---|---|---|---|---|---|---|---|---|---|
| | Baseline | After 4-week | After 8-week | Δ [4-week −baseline] | p*-value | Δ [8-week − 4-week] | p**-value | p***-value | Baseline | After 4-week | After 8-week | Δ [4-week −baseline] | p*-value | Δ [8-week − 4-week] | p**-value | p***-value |
| OSI | 1.06±0.43 | 0.47±0.15 | 0.6±0.25 | −0.59 | 0.001 | 0.13 | 0.001 | 0.88 | 0.8±0.29 | 0.8±0.29 | 0.64±0.17 | 0.0 | 1.0 | −0.16 | 0.21 | 0.51 |
| APSI | 0.73±0.42 | 0.29±0.13 | 0.46±0.24 | −0.44 | 0.001 | 0.17 | 0.001 | 0.02 | 0.59±0.20 | 0.56±0.19 | 0.45±0.15 | −0.03 | 1.0 | −0.11 | 0.13 | 0.51 |
| MLSI | 0.51±0.31 | 0.28±0.15 | 0.37±0.19 | −0.23 | 0.001 | 0.09 | 0.04 | 0.52 | 0.46±0.24 | 0.45±0.22 | 0.45±0.21 | −0.01 | 1.0 | 0.0 | 1.0 | 1.0 |

Significant differences *- within-group differences between 4 weeks after exercise and baseline, **- within-group differences between 8 weeks after (4 weeks after intervention) and baseline; ***- between-group differences between 8 weeks and 4 weeks; ($p \leq 0.05$).

**Table 4. Dominant leg on an unstable platform in the range 4 platform mobility level.**

| Vari-ables | Nordic Exercise training group | | | | | | | | Control group | | | | | | | |
|---|---|---|---|---|---|---|---|---|---|---|---|---|---|---|---|---|
| | Baseline | After 4-week | After 8-week | Δ [4-week −baseline] | p*-value | Δ [8-week − 4-week] | p**-value | p***-value | Baseline | After 4-week | After 8-week | Δ [4-week −baseline] | p*-value | Δ [8-week − 4-week] | p**-value | p***-value |
| OSI | 1.22±0.53 | 0.71±0.29 | 0.79±0.33 | −0.51 | 0.001 | 0.08 | 0.001 | 1.0 | 1.31±0.34 | 1.15±0.33 | 1.13±0.35 | −0.16 | 0.24 | −0.02 | 0.07 | 1.0 |
| APSI | 0.83±0.44 | 0.42±0.20 | 0.48±0.20 | −0.41 | 0.001 | 0.06 | 0.001 | 1.0 | 0.84±0.28 | 0.73±0.25 | 0.66±0.21 | −0.11 | 0.9 | −0.07 | 0.05 | 1.0 |
| MLSI | 0.68±0.27 | 0.33±0.10 | 0.45±0.20 | −0.35 | 0.001 | 0.12 | 0.001 | 0.13 | 0.67±0.20 | 0.55±0.18 | 0.53±0.19 | −0.12 | 0.2 | −0.02 | 0.04 | 1.0 |

Significant differences *- within-group differences between 4 weeks after exercise and baseline, **- within-group differences between 8 weeks after (4 weeks after intervention) and baseline; ***- between-group differences between 8 weeks and 4 weeks; ($p \leq 0.05$).

**Table 5. Mean±SD OSI, APSI, and MLSI between the NHE group and the CON group.**

| Unstable platform | Time | NHE Group Mean±SD | CON Group Mean±SD | P value* |
|---|---|---|---|---|
| OSI | Baseline | 1.06±0.43 | 0.8±0.29 | 0.24 |
| | After 4-week | 0.47±0.15 | 0.8±0.29 | 0.03 |
| | After 8-week | 0.60±0.25 | 0.6±0.17 | 1.0 |
| APSI | Baseline | 0.73±0.42 | 0.59±0.20 | 1.0 |
| | After 4-week | 0.29±0.13 | 0.56±0.19 | 0.04 |
| | After 8-week | 0.46±0.24 | 0.45±0.15 | 1.0 |
| MLSI | Baseline | 0.51±0.31 | 0.46±0.24 | 1.0 |
| | After 4-week | 0.28±0.15 | 0.45±0.22 | 0.66 |
| | After 8-week | 0.37±0.19 | 0.45±0.21 | 1.0 |
| **Dominant leg on an unstable platform** | **Time** | **NHE Group Mean±SD** | **CON Group Mean±SD** | |
| OSI | Baseline | 1.22±0.53 | 1.31±0.34 | 1.0 |
| | After 4-week | 0.71±0.29 | 1.15±0.33 | 0.02 |
| | After 8-week | 0.79±0.33 | 1.13±0.35 | 0.24 |
| APSI | Baseline | 0.83±0.44 | 0.84±0.28 | 1.0 |
| | After 4-week | 0.42±0.20 | 0.73±0.25 | 0.04 |
| | After 8-week | 0.48±0.20 | 0.66±0.21 | 1.0 |
| MLSI | Baseline | 0.68±0.27 | 0.67±0.20 | 1.0 |
| | After 4-week | 0.33±0.10 | 0.55±0.18 | 0.04 |
| | After 8-week | 0.45±0.20 | 0.53±0.19 | 1.0 |

Significant differences *- between the NHE group and the CON group; (p≤0.05).

For the left leg, two-way mixed-model ANOVAs revealed significant main effects of *Time* for knee flexors (F(2,58) = 26.2, p<0.001, η²=0.47) and extensors (F(2,60) = 54.5, p<0.001, η²=0.64). Post hoc analyses showed that concentric knee flexor and extensor forces significantly increased at 4 and 8 weeks compared to baseline in the NHE group (p≤0.001). In the CON group, significant improvements were observed at 4 and 8 weeks for knee extensors (p≤0.001) and at 4 and 8 weeks for knee flexors (p=0.04 and p=0.02, respectively) (Fig 3).

## Discussion

This study investigated the effects of a 4-week NHE training program on postural balance and muscle strength, marking a significant advancement in understanding the multifaceted benefits of NHE. Notably, this is the first study to assess postural balance using three indices: Overall Stability Index (OSI), Anterior-Posterior Stability Index (APSI), and Mediolateral Stability Index (MLSI) on both stable and unstable platforms. The results demonstrated a marked improvement in postural balance following the NHE training, supporting previous findings that NHE can enhance dynamic balance in athletes, thereby potentially reducing injury risk and associated healthcare costs.

In line with previous research, such as that by Anastasi and Hamzeh [19], which highlighted the effectiveness of a 10-week NHE program in reducing bilateral muscle strength imbalance and improving vertical jump height, our findings further substantiate NHE's role as a preventive and rehabilitative tool for hamstring injuries. Additionally, review by Nunes et al. [20] corroborates these results, indicating that NHE interventions can lead to significant reductions in hamstring injuries—up to 51%—while also enhancing sprint performance and muscle activation. The innovative aspect of this study lies in its comprehensive assessment of postural balance through multiple indices, providing a deeper understanding of how NHE influences not only hamstring strength but also overall stability. This approach is in line with

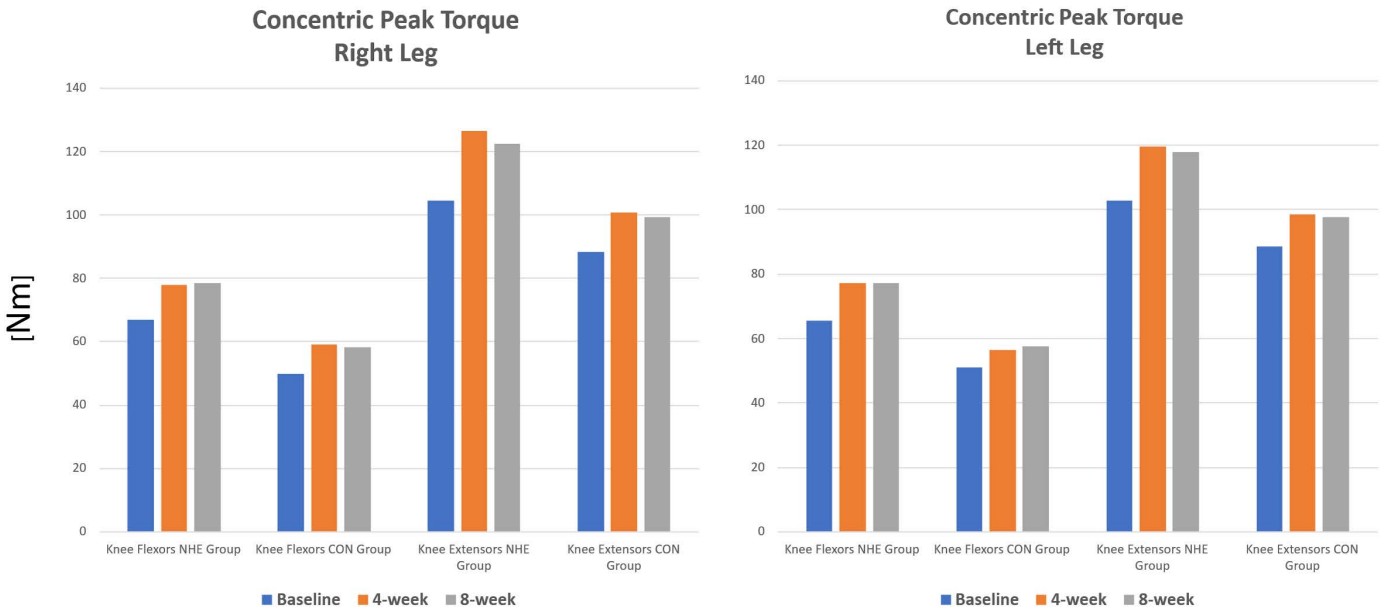

**Fig 3. Mean±SD Concentric peak torque for right leg and left leg.**

from Clark et al. [21], who noted that NHE improves neuromuscular adjustments crucial for injury prevention. Furthermore, our findings emphasize the importance of core muscle co-activation during NHE, as highlighted by Narouei et al. [22], suggesting that effective injury prevention strategies should incorporate exercises targeting both the hamstrings and core muscles.

This study investigated changes in postural balance and muscle strength after a 4-week NHE training program. To the best of the author's knowledge, this is the first study to assess the impact of postural balance in three indices: OSI, APSI, and MLSI using a stable and unstable platform. The results of this study demonstrated an improvement in postural balance after 4 weeks of NHE training. The findings supported our hypotheses and may provide new insights into the effects of NHE on hamstring muscles. As Saleh et al. [7] suggested, NHE can significantly improve dynamic balance in athletes, potentially reducing injury risk and associated healthcare costs. Several studies [23–29] have demonstrated that implementing NHE in injury prevention programs can effectively reduce injury incidence. This study used a 4-week NHE training protocol similar to Alt et al. [13], consisting of three sessions per week with three sets of three repetitions per session. Alt et al. [13] showed that this protocol significantly strengthened hamstrings and improved muscle balance, contributing to hamstring injury prevention. Suskens et al. [30] and Elerian et al. [8] also highlighted the effectiveness of NHE in preventing hamstring injuries, particularly when combined with other exercises or used as a pre-and post-training intervention. Clark et al. [21] demonstrated that NHE can improve neuromuscular adjustments, reducing injury risk factors. Narouei et al. [22] found that NHE activates not only hamstrings but also spinal extensors and internal oblique muscles, emphasizing the importance of core muscle co-activation in preventing injuries and improving postural balance. This aligns with our findings of improved postural balance after NHE training.

The NHE group showed significant improvements in OSI, APSI, and MLSI on the stable platform with closed eyes at both 4 and 8 weeks post-intervention compared to baseline. The control group showed no significant changes in any index. The NHE group also showed significant improvements in OSI, APSI, and MLSI on the unstable platform at both 4 and 8 weeks. Additionally, the NHE group had significantly better OSI and APSI scores than the control group at 4 weeks. Similar results were observed for the dominant leg on the unstable platform, with the NHE group showing significant

improvements in all three indices compared to the control group. Daneshjoo et al. [31] found that the FIFA+ and HarmoKnee programs, which include NHE, improved static and dynamic balance in professional male soccer players. Static balance with eyes closed improved by 10.9% in the FIFA+ group and 6.1% in the HarmoKnee group, while dynamic balance (measured by SEBT) improved by 12.4% in the FIFA+ group and 17.6% in the HarmoKnee group. Saleh et al. [7] demonstrated that 6 weeks of NHE and Copenhagen adduction exercises significantly improved dynamic balance, as measured by the limit of stability (LoS) on the Biodex Stability System, in amateur male athletes. Leavey et al. [32] reported that 6 weeks of combined balance and strength training improved dynamic postural control, as evaluated by the SEBT, in healthy males and females.

As Pickerill et al. [33] noted, proper postural balance, the ability to control the body's center of gravity, is essential for daily activities and athletic performance. NHE activates posterior hip and trunk muscles, crucial for maintaining postural balance, as shown by Narouei and Willson [22,34]. Long-term muscle imbalances can increase the risk of recurrent hamstring injuries, as suggested by Croisier et al. [35,36]. Isokinetic muscle strength tests were performed at baseline, 4 weeks, and 8 weeks. Both the NHE and control groups showed increased knee flexor and extensor strength in both legs. Daneshjoo et al. [37] also found increased muscle strength in both dominant and non-dominant legs after a 24-session program. Eccentric exercise, as suggested by Chen et al. [38], can improve neural control and knee muscle strength, potentially altering muscle balance at the knee joint. Eom et al. [39] found that changing the knee joint angle from flexion to extension increases trunk muscle activity. Improved balance control is crucial for athletes, who frequently challenge their balance dynamically during activities like sprinting and jumping [7,40].

## Practical implications

This study suggests that incorporating eccentric NHE exercises into training programs for amateur and professional athletes can improve postural balance by activating the posterior thigh muscles. This may ultimately reduce the risk of hamstring injuries and the need for subsequent rehabilitation, which often includes eccentric exercises.

## Limitations

This study has some potential limitations. First, it was conducted solely with university soccer athletes. Future research should encompass a broader range of participants, including professional athletes in both team and individual sports, to investigate potential group-dependent differences in static and dynamic balance. Second, this study did not analyze inter-gender differences in static and dynamic balance. Future studies should focus on changes in static and dynamic balance between males and females. Third, future research should consider a more homogeneous study group, including both injured and healthy athletes, to gain deeper insights into the effects of NHE on postural balance. Finally, extending the follow-up period beyond 4 weeks would provide insights into the long-term effects of NHE on postural balance.

## Conclusion

In conclusion, this study demonstrated that a 4-week NHE training program significantly improved static and dynamic balance, as measured by OSI, APSI, and MLSI, in university soccer athletes. These improvements were sustained 4 weeks post-training, while the control group showed no significant changes. These findings highlight the effectiveness of NHE in enhancing postural balance and performance in this population. Future research should investigate the effects of NHE on a broader range of athletes, including professionals and those in individual sports, to determine the generalizability of these findings. Furthermore, exploring the impact of NHE on injury prevention and athletic performance outcomes, such as speed, agility, and reaction time, would provide valuable insights. Monitoring postural control through assessments like OSI, APSI, and MLSI is crucial for evaluating neuromuscular function and optimizing training programs for injury prevention in athletes.

## Supporting information

**S1 File. Study protocol.**
(DOC)

**S2 File. CONSORT 2010 checklist.**
(DOC)

## Author contributions

**Conceptualization:** Magdalena Podczarska-Głowacka, Ewelina Perzanowska, Katarzyna Krasowska.

**Data curation:** Magdalena Podczarska-Głowacka, Ewelina Perzanowska, Zuzanna Trapik.

**Formal analysis:** Magdalena Podczarska-Głowacka.

**Investigation:** Magdalena Podczarska-Głowacka, Agata Kalkowska.

**Methodology:** Magdalena Podczarska-Głowacka, Ewelina Perzanowska, Katarzyna Krasowska, Zuzanna Trapik, Agata Kalkowska.

**Project administration:** Magdalena Podczarska-Głowacka, Sebastian Klich.

**Writing – original draft:** Magdalena Podczarska-Głowacka, Katarzyna Krasowska.

**Writing – review & editing:** Magdalena Podczarska-Głowacka, Sebastian Klich.

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
