## [Decision Letter · Decision Letter 0]

Dear Dr. Podczarska-Głowacka,

Thank you for submitting your manuscript to PLOS ONE. After careful consideration, we feel that it has merit but does not fully meet PLOS ONE’s publication criteria as it currently stands. Therefore, we invite you to submit a revised version of the manuscript that addresses the points raised during the review process.

We look forward to receiving your revised manuscript.

Kind regards,

Julio Alejandro Henriques Castro da Costa

Academic Editor

PLOS ONE

Journal Requirements:

2. We note that you have selected “Clinical Trial” as your article type. PLOS ONE requires that all clinical trials are registered in an appropriate registry (the WHO list of approved registries is at https://www.who.int/clinical-trials-registry-platform/network/primary-registries " https://www.who.int/clinical-trials-registry-platform/network/primary-registries and more information on trial registration is at http://www.icmje.org/about-icmje/faqs/clinical-trials-registration/ ). Please state the name of the registry and the registration number (e.g. ISRCTN or ClinicalTrials.gov) in the submission data and on the title page of your manuscript. a) Please provide the complete date range for participant recruitment and follow-up in the methods section of your manuscript. b) If you have not yet registered your trial in an appropriate registry, we now require you to do so and will need confirmation of the trial registry number before we can pass your paper to the next stage of review. Please include in the Methods section of your paper your reasons for not registering this study before enrolment of participants started. Please confirm that all related trials are registered by stating: “The authors confirm that all ongoing and related trials for this drug/intervention are registered”. Please see http://journals.plos.org/plosone/s/submission-guidelines#loc-clinical-trials for our policies on clinical trials.

3. Please include captions for your Supporting Information files at the end of your manuscript, and update any in-text citations to match accordingly. Please see our Supporting Information guidelines for more information: http://journals.plos.org/plosone/s/supporting-information .

Reviewers' comments:

Reviewer's Responses to Questions

**Comments to the Author**

1. Is the manuscript technically sound, and do the data support the conclusions?

Reviewer #1: Yes

Reviewer #2: Yes

Reviewer #3: Partly

2. Has the statistical analysis been performed appropriately and rigorously?

Reviewer #1: Yes

Reviewer #2: Yes

Reviewer #3: No

3. Have the authors made all data underlying the findings in their manuscript fully available?

Reviewer #1: Yes

Reviewer #2: No

Reviewer #3: Yes

4. Is the manuscript presented in an intelligible fashion and written in standard English?

Reviewer #1: Yes

Reviewer #2: No

Reviewer #3: Yes

Reviewer #1: Thank you for submitting your manuscript, which provides valuable insights into the effects of Nordic Hamstring Exercises on postural balance and muscle strength. Overall, the study is well-structured and contributes significantly to the field. However, there are a few areas where improvements can enhance the clarity and impact of your work.

In the abstract, adding practical implications and briefly mentioning key methodological details such as sample size and statistical significance would strengthen its completeness. In the introduction, highlighting the innovative aspects of your study and better contextualizing the findings within the existing literature would further support the relevance of your work. The methods section could benefit from additional details regarding group balancing, the monitoring of exercise protocols, and a more explicit justification of the statistical methods used.

In the results section, statistical significance could be emphasized more clearly in the tables, and the clinical relevance of the findings could be elaborated. The discussion would be strengthened by a more detailed examination of the study’s limitations, the generalizability of the findings, and suggestions for future research on the long-term effects of Nordic Hamstring Exercises. The conclusion could also expand on future research directions and discuss the broader implications of the findings on sports performance.

Finally, the figures and tables could be made more user-friendly by improving visual elements such as scales and coloring, and adding explanatory notes. Ensuring consistency in reference formatting and providing more detailed explanations for methodological references would further improve the manuscript’s readability and coherence. Addressing these points will help enhance the overall quality and impact of your manuscript.

Reviewer #2: General Overview

Your research on how a 4-week Nordic Hamstring Exercise (NHE) program affects postural balance is timely and pertinent, considering how common hamstring injuries are in sports. The Biodex Balance System and the randomised controlled trial (RCT) design are noteworthy advantages, providing accurate and unbiased assessments of postural balance. The manuscript has to be significantly revised, nevertheless, in order to improve its organisation, clarity, and level of analysis.

Specific Comments:

1. Introduction

• The introduction highlights the importance of hamstring injury prevention and NHE, but the connection between NHE and postural balance could be more clearly articulated earlier.

• A clearly defined research question or hypothesis would improve the focus and direction of the study.

2. Methodology

• The use of two-way repeated measures ANOVA and Pearson correlation is appropriate, but further explanation of participant retention rates is needed. Were there any dropouts? How were they handled in the analysis?

• Consider adding a gender-based analysis to explore potential differences in outcomes.

• Extending the follow-up period beyond 4 weeks would provide insights into the long-term effects of NHE on postural balance.

3. Results

• The results section is thorough, but more emphasis on gender-specific differences and their implications would enhance the analysis.

• Clarify how retention rates or any missing data might have influenced the findings.

4. Discussion

• The discussion effectively integrates your findings with previous research, but a more explicit identification of research gaps would strengthen the narrative.

• Limitations are acknowledged, but consider elaborating on the impact of sample size, participant diversity, and previous injury history.

5. Bibliography/References

• The references are relevant and comprehensive, but a tighter focus on NHE’s direct impacts on muscle balance, strength, and injury prevention would strengthen the evidence base. Ensure that all cited studies are directly related to the core research question.

6. Language and Presentation

• Some sentences are dense and complex. Simplifying these will improve readability.

• Grammar and flow require attention. Consider a professional language editing service if necessary.

Ethical Considerations

There are no apparent concerns regarding research or publication ethics. However, please ensure data availability and transparency for reproducibility.

Recommendation

Requires Major Revision

Your manuscript's clarity, rigour, and impact will all be greatly enhanced by following these suggestions.

Reviewer #3: The primary aim of this study is to compare the treatment group with the placebo group. However, the primary endpoint was not met, as no significant difference was observed between the groups. This point needs to be clarified in both the abstract and the conclusion.

Further clarification is needed regarding the sample size. Was the sample size calculated specifically for between-group comparisons? What effect size was used for the calculation? Was it based on the between-group difference at 8 weeks or across all time points? Additionally, what is the beta? The chosen effect size and beta value must be clinically justified.

On page 11, it would be more appropriate to move Table 1 and the participant information to the Results section.

For Tables 2–4, it is unclear whether the p-values from the post-hoc analyses were adjusted. The authors also mention “p = 0.01” in the statistical methods section, suggesting the use of raw p-values but with a stricter cutoff. This requires clarification.

The column p-value *** in Tables 2–4 are ambiguous. The footnote says “***- between-group differences between 8 weeks and 4 weeks; (p≤0.05).” . If I understand correctly, there should be two p-values for each outcome—one for the 8-week comparison and one for the 4-week comparison between NHE and CON. However, only a single p-value is presented for each outcome, which needs further explanation. What does "(p≤0.05)" mean here?

Additionally, the purpose of the correlation analysis is unclear and appears to lack relevance to the primary aim of the clinical trial.

**Do you want your identity to be public for this peer review?** For information about this choice, including consent withdrawal, please see our Privacy Policy

Reviewer #1: **Yes: ** GÖKHAN DELİCEOĞLU

Reviewer #2: **Yes: ** D. Kalidoss

Reviewer #3: No

---

## [Author Response · Author response to Decision Letter 1]

19 Mar 2025

Ref ID: PONE-D-24-47476

Title: The influence of 4-week eccentric Nordic Hamstring Exercise training on postural balance and muscle strength: a randomized controlled trial

Journal: PLOS ONE

Answers to the Reviewer #1 (all correction and changes are marked with grey). Thank you for your review and opinions on our manuscript that have contribute to increase its overall quality. We appreciate your efforts in the review process.

Reviewer #1: In the abstract, adding practical implications and briefly mentioning key methodological details such as sample size and statistical significance would strengthen its completeness.

Answer: Thank you for your detailed comment and suggestion.

Action: We improved the abstract according to your suggestions adding practical implications and briefly mentioning key methodological details such as sample size and statistical significance.

1. We added practical implications at the end of the conclusions.

“This study suggests that eccentric NHE exercises may improve athlete balance, reduce hamstring injury risk, and decrease the need for rehabilitation by activating posterior thigh muscles.”

2. We clarified methodological details.

“This study involved thirty-two recreationally active individuals, randomly divided into two groups: Nordic Exercise training (NHE Group; n=16) and the matched-control group (CON Group; n=16). The NHE Group performed Nordic hamstring exercise training for 4 weeks, during which the volunteer exercised 3 times a week for 3 sets, each set consisting of 3 repetitions of NHE.”

3. We added p-value in the results.

“The NHE group significantly improved balance on a stable platform with eyes closed compared to baseline (p ≤ 0.001) and on unstable platforms compared to both baseline and the control group (p ≤ 0.001). Both groups improved knee strength, but the NHE group showed significantly greater improvements (p ≤ 0.001 for NHE, p ≤ 0.04 for control). Strong positive correlations (r = 0.5 to 0.7) were found between stability measures, suggesting shared underlying mechanisms influencing balance (p < 0.01).”

Reviewer #1: In the introduction, highlighting the innovative aspects of your study and better contextualizing the findings within the existing literature would further support the relevance of your work.

Action: We added a research perspective to the introduction.

“A decreased level of muscular strength and postural balance has been related to a higher risk of hamstring muscle injuries [1,2]. From a biomechanical perspective, assessing the interaction between muscular strength, balance, and injury risk is complex. These factors are interconnected and influenced by various biomechanical and neuromuscular properties of the musculoskeletal system [3, 4]. Previous studies investigated limit of stability after 6 weeks of NHE training using the Biodex Stability System [3], while other studies assessed dynamic balance after both single [5] and 6 weeks of NHE training using the Y-Balance Test. To our knowledge, the effectiveness of NHE training on postural balance remains unclear. Therefore, this study aimed to address the limitations of previous studies [3-5] by comprehensively assessing postural balance across three planes of motion (anterior-posterior, medial-lateral, and overall stability index ((APSI, MLSI, and OSI)), providing a more holistic understanding of the impact of NHE on balance. This innovative approach, by examining balance across multiple planes, may lead to a more nuanced understanding of the specific mechanisms by which NHE training improves balance and its implications for injury prevention.”

Reviewer #1: The methods section could benefit from additional details regarding group balancing, the monitoring of exercise protocols, and a more explicit justification of the statistical methods used.

Action: We added following updates:

To study design: “This study employed a randomized, controlled, single-blind design with a repeated-measures approach. To ensure rigorous reporting, we adhered to the Consolidated Standards of Reporting Trials (CONSORT) guidelines for pragmatic trials. Group allocation was randomized, resulting in two equal groups of 16 participants, including Nordic Exercise training (NHE) and a control (CON) group.”

To the group allocation: “The recruitment process consisted of an interview during which questions were asked about physical activity, training experience, and weekly duration of activity. Participants were randomly divided into two groups, i.e.: NHE and CON. Allocation was determined using a random number generator (random.org). Each participant was assigned a unique number, corresponding to a sealed envelope containing their group assignment. An independent researcher, blinded to the group assignments, selected these envelopes. Participants learned their group assignment only after completing baseline data collection. This information remained confidential from the researcher conducting the study.”

To the monitoring of exercise protocols: “A week before the start of the experimental procedure, participants were invited to the Physical Exercise Laboratory of AWFiS in Gdańsk to familiarize themselves with the Nordic Hamstring exercise procedure and all outcome devices. All participants were familiarized with the research procedure and all test procedures during the measurement conducted 7 days before the main test. During the experiment, all participants completed a standardized warm-up routine. This included a warm-up on a cycle ergometer for 15 minutes, at 40 W and a speed of 20 km/h [11]. After completing baseline measurements, the NHE group received the intervention, while the CON group received no intervention.”

To statistical analysis: “Normality of data distribution was assessed using the Shapiro-Wilk test, while homogeneity of variance was evaluated with Levene's test. A two-way 3 x 2 mixed-model analysis of variance (ANOVA) with Time (baseline, after 4 weeks, and after 8 weeks) and Intervention (NHE group and CON group) was analyzed for the Postural Stability Test.”

Reviewer #1: In the results section, statistical significance could be emphasized more clearly in the tables, and the clinical relevance of the findings could be elaborated.

Answer: Thank you for this suggestion. In our opinion, the tables might be more readable.

Action: We re-arrange the tables.

Reviewer #1: The discussion would be strengthened by a more detailed examination of the study’s limitations, the generalizability of the findings, and suggestions for future research on the long-term effects of Nordic Hamstring Exercises.

Action: We added to the limitations following statements: “This study has some potential limitations. First, it was conducted solely with university soccer athletes. Future research should encompass a broader range of participants, including professional athletes in both team and individual sports, to investigate potential group-dependent differences in static and dynamic balance. Second, this study did not analyze inter-gender differences in static and dynamic balance. Future studies should focus on changes in static and dynamic balance between males and females. Third, future research should consider a more homogeneous study group, including both injured and healthy athletes, to gain deeper insights into the effects of NHE on postural balance. Finally, extending the follow-up period beyond 4 weeks would provide insights into the long-term effects of NHE on postural balance.”

Reviewer #1: The conclusion could also expand on future research directions and discuss the broader implications of the findings on sports performance.

Action: We updated the conclusions with following sentences “In conclusion, this study demonstrated that a 4-week NHE training program significantly improved static and dynamic balance, as measured by OSI, APSI, and MLSI, in university soccer athletes. These improvements were sustained 4 weeks post-training, while the control group showed no significant changes. These findings highlight the effectiveness of NHE in enhancing postural balance and performance in this population. Future research should investigate the effects of NHE on a broader range of athletes, including professionals and those in individual sports, to determine the generalizability of these findings. Furthermore, exploring the impact of NHE on injury prevention and athletic performance outcomes, such as speed, agility, and reaction time, would provide valuable insights. Monitoring postural control through assessments like OSI, APSI, and MLSI is crucial for evaluating neuromuscular function and optimizing training programs for injury prevention in athletes.”

Reviewer #1: Finally, the figures and tables could be made more user-friendly by improving visual elements such as scales and coloring, and adding explanatory notes. Ensuring consistency in reference formatting and providing more detailed explanations for methodological references would further improve the manuscript’s readability and coherence. Addressing these points will help enhance the overall quality and impact of your manuscript.

Answer: We agree, that the current figures are not clear and prepared properly.

Action: According to your previous comment, we re-arranged and re-edit all tables.

Ref ID: PONE-D-24-47476

Title: The influence of 4-week eccentric Nordic Hamstring Exercise training on postural balance and muscle strength: a randomized controlled trial

Journal: PLOS ONE

Answers to the Reviewer #2 (all correction and changes are marked with grey). Thank you for your review and opinions on our manuscript that have contribute to increase its overall quality. We appreciate your efforts in the review process.

Reviewer #2: Your research on how a 4-week Nordic Hamstring Exercise (NHE) program affects postural balance is timely and pertinent, considering how common hamstring injuries are in sports. The Biodex Balance System and the randomised controlled trial (RCT) design are noteworthy advantages, providing accurate and unbiased assessments of postural balance. The manuscript has to be significantly revised, nevertheless, in order to improve its organisation, clarity, and level of analysis..

Answer: Thank you for your opinion and suggestions. According to your and other reviewers' comments, we re-edited and re-organized the manuscript.

Reviewer #2: Introduction. The introduction highlights the importance of hamstring injury prevention and NHE, but the connection between NHE and postural balance could be more clearly articulated earlier.

Action: We added a following information in the introduction.

“The connection between NHE and postural balance is further supported by research indicating that NHE can significantly enhance dynamic balance performance among athletes. A study found that participants who performed NHE demonstrated improved balance capabilities, which are essential for reducing injury risks and enhancing overall motor performance in sports settings (Babu et al., 2018).”

Reviewer #2: Introduction. A clearly defined research question or hypothesis would improve the focus and direction of the study.

Action: We added a hypothesis “We hypothesized that a 4-week NHE eccentric training program will significantly improve dynamic balance in recreationally active individuals compared to a control group. This improvement will be observed across all three planes of postural stability (APSI, MLSI, and OSI) and in hamstring muscle strength.”

Reviewer #2: Methodology. The use of two-way repeated measures ANOVA and Pearson correlation is appropriate, but further explanation of participant retention rates is needed. Were there any dropouts? How were they handled in the analysis?

Answer: Thank you for this comment. This explanation might be very important to clarify the methodology and sample size. First, we re-edited the statistical analysis. A two-way 3 x 2 mixed-model analysis of variance (ANOVA) with Time (baseline, after 4 weeks, and after 8 weeks) and Intervention (NHE group and CON group) was analyzed for the Postural Stability Test. From the statistical point of view, a mixed model is more appreciated in order of missing data. Second, we re-edit the CONSORT 2010 Diagram Flow including Enrolment, randomization, and drop out of participants allocated to the intervention group (NHE) and matched-control group (CON). In our study, participants were not excluded from enrollment and further analysis.

Reviewer #2: Methodology. Consider adding a gender-based analysis to explore potential differences in outcomes.

Answer: The total sample size is to small to perform an additional gender-based analysis.

The total gender distribution is as follows, i.e., men (n=13) and women (n=19).

Action: According to lack of proper distribution of the number of men and women, we included this analysis in the limitations and directions for future research in the discussion.

Reviewer #2: Methodology. Extending the follow-up period beyond 4 weeks would provide insights into the long-term effects of NHE on postural balance.

Answer: The primary focus of this study was to investigate the immediate effects of a 4-week NHE training program on dynamic balance and muscle strength. Extending the follow-up period would have shifted the focus and potentially introduced confounding variables unrelated to the primary objective. Practical limitations, such as participant availability and the potential for confounding factors like changes in regular physical activity outside the study, may have made it difficult to maintain participant engagement and ensure data reliability in a longer follow-up period.

Answer: We uploaded the limitations.

“This study has some potential limitations. First, it was conducted solely with university soccer athletes. Future research should encompass a broader range of participants, including professional athletes in both team and individual sports, to investigate potential group-dependent differences in static and dynamic balance. Second, this study did not analyze inter-gender differences in static and dynamic balance. Future studies should focus on changes in static and dynamic balance between males and females. Third, future research should consider a more homogeneous study group, including both injured and healthy athletes, to gain deeper insights into the effects of NHE on postural balance. Finally, extending the follow-up period beyond 4 weeks would provide insights into the long-term effects of NHE on postural balance.”

Reviewer #2: Results. The results section is thorough, but more emphasis on gender-specific differences and their implications would enhance the analysis.

Answer: We agree with your suggestion. As you previously commented, we have added the lack of this analysis to the limitations and directions for future research in the discussion section.

Reviewer #2: Results. Clarify how retention rates or any missing data might have influenced the findings.

Answer: In order of any missing data we re-calculated the analysis to mixed-model ANOVA.

Reviewer #2: Discussion. The discussion effectively integrates your findings with previous research, but a more explicit identification of research gaps would strengthen the narrative.

Action: Thank you for this important suggestion. We re-edited this section.

“In line with previous research, such as that by Anastasi and Hamzeh (2011), which highlighted the effectiveness of a 10-week NHE program in reducing bilateral muscle strength imbalance and improving vertical jump height, our findings further substantiate NHE's role as a preventive and rehabilitative tool for hamstring injuries. Additionally, review by Nunes et al. (2024) corroborates these results, indicating that NHE interventions can lead to significant reductions in hamstring injuries—up to 51%—while also enhancing sprint performance and muscle activation. The innovative aspect of this study lies in its comprehensive assessment of postural balance through multiple indices, providing a deeper understanding of how NHE influences not only hamstring strength but also overall stability. This approach is in line with from Clark et al. (2020), who noted that NHE improves neuromuscular adjustments crucial for injury prevention. Fu

---

## [Decision Letter · Decision Letter 1]

The influence of 4-week eccentric Nordic Hamstring Exercise training on postural balance and muscle strength: a randomized controlled trial

PONE-D-24-47476R1

Dear Dr. Podczarska-Głowacka,

We’re pleased to inform you that your manuscript has been judged scientifically suitable for publication and will be formally accepted for publication once it meets all outstanding technical requirements.

Kind regards,

Julio Alejandro Henriques Castro da Costa

Academic Editor

PLOS ONE

Additional Editor Comments (optional):

Reviewers' comments:

Reviewer's Responses to Questions

**Comments to the Author**

Reviewer #3: All comments have been addressed

2. Is the manuscript technically sound, and do the data support the conclusions?

Reviewer #3: (No Response)

3. Has the statistical analysis been performed appropriately and rigorously?

Reviewer #3: (No Response)

4. Have the authors made all data underlying the findings in their manuscript fully available?

Reviewer #3: (No Response)

5. Is the manuscript presented in an intelligible fashion and written in standard English?

Reviewer #3: (No Response)

Reviewer #3: All my concerns are addressed.

**Do you want your identity to be public for this peer review?** For information about this choice, including consent withdrawal, please see our Privacy Policy

Reviewer #3: No

---

## [Editor Report · Acceptance letter]

PONE-D-24-47476R1

PLOS ONE

Dear Dr. Podczarska-Głowacka,

I'm pleased to inform you that your manuscript has been deemed suitable for publication in PLOS ONE. Congratulations! Your manuscript is now being handed over to our production team.

Kind regards,

on behalf of

Dr. Julio Alejandro Henriques Castro da Costa

Academic Editor

PLOS ONE